# Research on the Recognition Performance of Bionic Sensors Based on Active Electrolocation for Different Materials

**DOI:** 10.3390/s20164608

**Published:** 2020-08-17

**Authors:** Wenhao Du, Yu’e Yang, Luning Liu

**Affiliations:** School of Mechanical Engineering, University of Jinan, Jinan 250022, China; 201821200559@mail.ujn.edu.cn (W.D.); me_liuln@ujn.edu.cn (L.L.)

**Keywords:** bionic sensor, active electrolocation, identification, lift-off distance

## Abstract

Underwater object identification by optical sensors is usually difficult in turbid or dark environments. The objective of this paper was to identify different underwater materials using active electrolocation technology. We proposed a bionic sensor inspired by the weakly electric fish. The material identification was completed by analyzing electric signal images, since the electric signal changes when different materials are identified. Firstly, the effective lift-off distance for identification was researched. The materials used in this paper can be effectively identified by the sensor at a lift-off distance of 10 mm. Furthermore, the performance of the sensor for identifying and locating was studied in the presence of multiple materials. The results indicated that the sensor can effectively identify and locate the objects when the distance between objects is greater than 30 mm, while the location error is less than 5% in most cases. Our research proves that the bionic sensor we made can effectively recognize different materials underwater in short-range, which is about 10 mm. Therefore, we expect that the bionic sensor we made can be utilized as a useful tool for underwater object identification.

## 1. Introduction

Since the “Nanhai No.1” shipwreck of the Song Dynasty was found in 2013, researchers have found iron nails, grains, and other items from the ship. When salvaging shipwrecks, a reasonable salvaging program needs to be developed according to the materials of the objects. But these objects’ material properties are diverse. After immersion and corrosion in the sea, it is difficult to identify the material of the objects solely by their appearance [1,2].

In waters where the depth is less than 50 m, frogmen are mainly used for checking and salvaging objects. When the water depth exceeds 50 m, it is dangerous for frogmen to work, so sonar sensors and underwater camera equipment are used. Sonar sensors are suitable for searching for objects, such as submarines, a shoal of fish, or a shipwreck. The useful range of the sensor is usually within 30 km, and its detection performance is limited by the high-level side lobes and seriously degraded in a shallow water environment due to time spread induced by multipath propagation. Therefore, underwater environments pose significant challenges in identifying targets underwater [3,4]. The images collected by underwater camera equipment are not satisfactory because of the impact of light in turbid water. Moreover, the color causes serious difficulty in underwater recognition [5]. Therefore, it is necessary to develop a new underwater recognition technology to make up for the shortage of existing sonar sensors and underwater camera equipment.

In turbid waters, there is a kind of fish that can send and receive electrical signals through its tissues and organs to detect the field changes in its surroundings. The fish employs electric pulses, of which the amplitude is about 2 V and the frequency is about 800 Hz, to explore their surroundings, realize underwater navigation, and communicate with other fish [6]. As the signal it sends out is extremely weak, it is called weakly electric fish [7]. Active electrolocation technology, which relies on low frequency electric fields, has attracted much attention in the field of biology and bionics in recent years. Many scholars have carried out research studies in the field of biology and bionics [8].

Machin and Lissmann established a mathematical model of object location in weakly electric fish named Gymnarchus Niloticus, and conducted multiple theoretical studies. The studies provided theoretical guidance for active electrolocation [9]. Philips studied the types of electric field signals emitted by different fish species. [10]. Fechler researched the ability of electrical signals from weakly electric fish to distinguish objects at different distances. The study showed that weakly electric fish could distinguish objects within a range of 3–4 cm [11]. Rother found the effect of the dielectric around the weak current fish on the skin surface voltage of the weak current fish, which provided the foundation for the application of active electrolocation [12]; von der Emde proposed the capacitance value of detectable objects of weakly electric fish. His experimental results show that the electrical signal characteristics generated by different types of objects in the process of electrolocation was different and identified the range of detectable target capacitance value of weakly electric fish [7]. Nelson and Migliaro et al. found that weakly electric fish perceive objects mainly by detecting characteristics such as energy intensity, direction, and time, and then judge objects; they assert that these fish’s detectional ability is directly related to the distance between them and the objects, and describe the electric field information around the weakly electric fish in the form of an infographic [13].

In the field of engineering, the active electrolocation technology has gained much attention and has played an important role in the field of biosensors. Gottwald proposed a biosensor based on active electrolocation for the detection and analysis of coronary artery disease; he used synthetic plaques for experiments, which could be reliably detected by the sensor [14,15]. Lebastard constructed a sensor based on active electrolocation technology to locate and estimate the size of small objects; the sensor showed navigational ability in the tank [16]. Bai presented an algorithm to estimate the shape, size, direction, and position of an oval object; the algorithm estimates the object size and length-to-width ratio with an accuracy of around 10%, which is useful for engineering applications [17]. Gottwald designed a biomimetic electric camera inspired by weakly fish for underwater inspection, using an ’electric outline’ to provide information resembling a target’s optical contour [18]. In 2020, Peng studied the amplitude information frequency characteristics for the multi-frequency excitation of underwater active electrolocation systems [19,20].

The abovementioned research works provide theoretical guidance for active electrolocation technology, and its application in some fields proves its feasibility for future applications. Different from sonar and camera, active electrolocation technology uses an electric signal as the exciting signal, it recognizes objects by sensing changes in conductivity. This technology has the advantages of high positioning accuracy in close ranges and without underwater imaging, but it still has some problems in terms of practicality—for example, the recognition performance of the technology for different materials needs to be improved.

Based on the existing research, we developed a bionic sensor and selects several materials to study the recognizable distance of the sensor towards the materials. We tried to establish the function relation between the height of recognition and the amplitude of the electrical signal change. Secondly, based on the sensor’s ability to identify and locate objects of different materials, the relationship between object spacing and identification performance was established, so as to improve the practicability of active electrolocation technology in practical engineering.

The rest of this paper is organized as follows. The theory of active electrolocation and the method of the research are introduced in Section 2. In this section, we explain the experimental equipment and materials which were used in this paper. In Section 3, the performance of the sensor we made is investigated and discussed. Finally, the conclusion is presented in Section 4.

## 2. Materials and Methods

### 2.1. Bionic Theory

The weakly electric fish generates an electric field around itself through its own electric organ discharge (EOD) in the turbid water. The fish can perceive the surrounding information by detecting the distortion of the electric field around itself. It can obtain information about the type and distance of the surrounding objects. In 1996, Basnow used mathematics to express the active electric field of weakly electric fish. He used a sphere of radius *a* placed in a uniform electric field E_0_ to generate a purely dipolar potential proportional to the field. For the simulations presented here, he assumed that the electric field was uniform and equal to the value measured at the object’s center. At the position *r* from the object center, the perturbation is given by:(1)$$\delta_{\varphi }\left(r\right){=E}_{0}\cdot r(\frac{a}{r}{)}^{3}\frac{\rho_{1}{-\rho }_{2}{+i\omega \rho }_{1}\rho_{2}{(\varepsilon }_{2}{-\varepsilon }_{1})}{{2\rho }_{2}{+\rho }_{1}{+i\omega \rho }_{1}\rho_{2}{(2\varepsilon }_{1}{+\varepsilon }_{2})}$$
where $\rho_{1}$ is the resistivity of the water, which is 5 kΩ·cm; $\varepsilon_{1}\;$is the dielectric constant of water, which is about 7.1 pF/cm; $\rho_{2}$ is the resistivity of the spherical object of radius *a*; $\varepsilon_{2}$ is the dielectric strength of the spherical object of radius *a*; and ω is the angular frequency of the unperturbed electric field; *r* = |*r*|; i = $\sqrt{-1}$. 

Therefore, the main parameters of active electrolocation include the conductivity of the environment and the conductivity of the measured object. In addition, the distance between the measured object and the detection system and the size of the measured object also affects it [21].

### 2.2. Bionic Sensor and Test Device

By abstracting and simplifying the weakly electric fish, the electric organ discharge (EOD) of the fish was replaced by two emission electrodes and the receiving organ of the fish was replaced by two receiving electrodes. The electrode tubes consisted of a PVC tube and a cylinder made of titanium at the bottom. The titanium was used as electrode because it has excellent corrosion resistance and electric conductivity. The image of the sensor is shown in Figure 1a.

Four electrode tubes were fixed with a plate to replace the body of the weakly electric fish. The EOD of the weakly electric fish is in the same straight line with the receiving organs. However, in the design of the bionic sensor, the receiving electrodes were placed in the offset position, because the electric field generated by the sensor electrodes in this layout is better, as our previous research showed.

In order to carry out the research, we built an active electrolocation test device in the laboratory as shown in Figure 2. The device consisted of a computer, motor controller, VibRunner, sensor, three axis ball screw, and a tank. The exciting signals generated by VibRunner passed through the wire to the sensor. The sensor sent the signal into the water. At the same time, the signal was collected by the sensor from the water. The analysis and processing of the signals were carried out on the VibRunner and transmitted to the computer.

A cuboid tank sized 1000 mm × 450 mm × 450 mm made of glass was used for conducting the tests. The tank was filled with fresh water until it hit 80 mm. The conductivity of the fresh water was about 0.005 Siemens/m. We added sand into the water to simulate real sea conditions as much as possible. The three-axis lead screw system was placed on the water tank. The sensor was attached to the tip of a stick which was fixed on the Z-axis of the three-axis lead screw. A coupling was used to connect the lead screw with the stepping motor. As such, the movement of the sensor could be controlled by the motor controller.

The control of the stepping motors was completed by the motor controller and PC software. We used the VibRunner from M + P which was made in Germany for outputting a sine signal as the exciting signal, the amplitude of which was 5 V and the frequency 1000 Hz. The analysis and processing of signals were completed by SO Analyzer software which matched VibRunner.

### 2.3. Materials and Experiment Design

Four kinds of materials were selected as the objects to be identified for the experiment; the properties of the materials are shown in Table 1. To control the variables, the size of the identified objects in each group of tests was consistent. The variables were only the electrical conductivity of the identified objects and the lift-off distance.

The identified object was placed at the center of the tank. The sensor was moved to the position shown in Figure 3a. The center of the sensor was kept at the same horizontal line in the center of the recognized object. When testing, the sensor was moved to the position where the bottom of the sensor was “H” away from the top of the identified object. The “H” was the lift-off distance, shown in Figure 4b.

After the beginning of the identification experiment, the X-axis motor of the three-axis ball screw started to rotate uniformly at the speed of 2 r/s (20 mm/s). To simplify the experiment, the sensor moved only along the X-axis. The sensor continuously sent out sine wave signals while receiving electrical signals from the water. During the test, the collected electrical signal was saved on the PC for processing. Then, the value of “H” was changed and the above process was repeated. H was set at three values: 10 mm, 20 mm, and 30 mm. The tests were repeated more than six times.

For further study, we carried out two groups of tests to identify different materials using the sensor. As shown in Figure 4, two identified objects were placed at the bottom of the water tank, where “X” was the distance of two objects. In the first group of tests, the comparative materials were the H13 and PMMA. H13 is a metal material while PMMA is a nonmetal material; in the second group, the comparative materials were the eggshell and the poached egg; the main component of eggshell is calcium carbonate, which can be considered an inorganic mineral, while the main component of boiled egg is protein, which can be considered organic.

In order to further verify the performance of the sensor, the H13, eggshell, and poached egg were used as the identified objects. We repeated the above tests, but this time keeping the distance between the identified objects random.

### 2.4. Data Processing

According to the moving distance of the sensor, the sampling time was set to 22 s. The time-domain signal was used for the analysis. The time-domain signal can reflect the change of the signal amplitude with time. In order to display the results more clearly, the electrical signal data was processed by Matlab, and the peak searching function was used for processing to obtain the final identification. The function was given by:Pks = findpeaks (date,‘minepeakdistance’,mph)(2)
where mph is the minimum number of intervals between two peaks.

The comparison of electric images is shown in Figure 5. 

## 3. Results and Discussion

### 3.1. The Effect of Lift-off Distance on Identification

Four kinds of materials were identified by the bionic sensor. The images of four materials are shown in Figure 6, where the *y-axis* in the figure represents the amplitude of the electrical signal, and the *x-axis* is the time.

We observed the following when analyzing the images:The overall upward shift in the image amplitude ranged when the lift-off distance increased. The amplitude of the electric images changed when the sensor approached to the identified objects.When identifying the H13, the signal was in a stable state at both ends. At about 12 s, the amplitude of the electrical signal decreased, and the degree of change became more obvious as the lift-off distance decreased. At the height of 10 mm, the amplitude of the change reached the maximum value of 23 mV.When identifying PMMA, the electrical signal image was completely different from that of the H13 at about 12 s. The amplitude of the electric image first decreased and then increased at 12 s, the maximum value of the increase was 6 mV when the lift-off distance was 10 mm.In the eggshell identification tests, the amplitude of the electrical signal image decreased and then increased when the lift-off distance was 10 mm. The amplitude of the electric image increased was about 10 mV, which was completely different from the image of the PMMA.In the poached egg identification tests, the electric image was completely different from the image of the eggshell. The amplitude of the electric image decreased about 5 mV when the lift-off distance was 10 mm.

The variation of the amplitude with lift-off distance of each test material is depicted as the curve shown in Figure 6f.

We drew the following conclusion after analyzing the images: Firstly, the variation rules are different when identifying different materials; the difference lay in the negative growth of the H13 and the poached egg, and the positive growth of the PMMA and the eggshell. This is because the metal, being electrically conductive, absorbed the signals sent by the transmitting electrode, so the signals received by the receiving electrode were weakened. As for non-conductive materials such as the eggshell and PMMA, the electric signal could not pass through the identified object. The strength of the electric field was high above the identified objects. Therefore, the amplitude of the electric signal increased when the sensor passed above the identified object. The electrical conductivity of the organic material was between metal and nonmetal, and the variation of the electrical signal was inclined towards the metal materials, but the amplitude of variation was small.Secondly, the amplitude of electric signal increased with the decrease of the lift-off distance when dissimilar materials were identified at the same position. The closer the sensor approached to the objects, the more obvious the change in the amplitude.Finally, each material had a limit of lift-off distance; in the tests, in order to avoid a collision between the sensor and the recognized target, the minimum of the lift-off distance was set as 10 mm. When identifying the H13, the change of amplitude was about 6 mV and lift-off distance was 30 mm, but this value was received at a lift-off distance of 10 mm when identifying the PMMA. As such, when identifying different materials, the corresponding lift-off distance should be set according to the properties of the materials.

### 3.2. Influence of Spacing between Objects on Identification Effect

The above study found that the curve of different materials presented different trends under the same lift-off distance. This attempted use the change rules of the signals to identify and locate objects.

In the identification and positioning experimental research, the change rules of the different materials were used to complete the identification of different materials. The position of the objects could be obtained indirectly by the electrical signal image.

The results of the first group tests are shown in Figure 7. When the H13 and PMMA fit together without spacing, that is when X ≤ 30 mm, only the electrical signal characteristics of the H13 appear in the image, but none of the electrical signal characteristics of PMMA.

At first, the two kinds of materials could not be effectively identified. This situation did not change until *X* = 30 mm. In the subsequent tests, the electrical signal characteristics of H13 and PMMA appeared in the electrical signal, at which point the identification of objects could be performed.

The distance of sensor movement was calculated by calculating the time when the electrical signal image changed, and then the position information was obtained by using Equation (3). The calculated results (*l*) were compared with the actual measured results using Equation (2). The results are shown in Table 2.
(3)$$\delta =\frac{l-X}{X}\times 100\%$$
(4)l=∆t·r·p
where *r* is the motor speed, *p* is the lead of kad-screw, and δ is tolerance.

The results of the second group were shown in Figure 8. The same method was used to obtain position information and then compared with the actual measurement results, which are shown in Table 3.

After conducting our research and analysis, we believe the positioning error mainly comes from the ∆*t*, when looking for the specific time of the trough and peak, there is some error in confirming the exact time because of the high frequency; the error can be further reduced if the time is more accurate.

### 3.3. Identification Test of Three Kinds of Materials

In order to determine whether this identification method can work with more than two kinds of materials, we tested with three materials using the same method previously described. The result is shown in Figure 9. In this image, the electrical signal image showed three characteristics simultaneously when the sensor passed over the objects but remained steady at other times. We can draw the conclusion that the sensor we made can identify different materials effectively.

## 4. Conclusions

In summary, a bionic sensor was developed simulating the sensor of the weakly electric fish. In order to studying the identification performance of the sensor, we carried out a battery of tests and can draw the following conclusions:The sensor we developed can complete the identification and positioning of objects effectively underwater. The identification and position of the objects were completed by analyzing the electric signal received by the sensor.The effect of identification was affected by the lift-off distance. As such, the lift-off distance should be adjusted according to the identified material. Generally, the useful identification range of the sensor is 10–20 mm when the frequency of the signal is 1000 Hz and the amplitude of the signal is 5 V.The sensor we made can identify both metal and nonmetal materials effectively. Also, it can identify the type of the materials even if the materials are both nonmetals. In our experiment, because the variation rules of the electric images were different when identifying different materials, the amplitude of the change was also different even though identified materials were both nonmetals. For example, the amplitude change of the PMMA is about 6 mV, while the amplitude change of the eggshell is about 10 mV. The materials were able to be identified by the change of the amplitude.We found that the positioning accuracy is related to the distance between the objects, and the higher the distance, the higher the positioning accuracy. In most cases, the positioning accuracy will remain below 5%.

Previous studies confirmed that a sensor based on active electrolocation could locate the objects in the water or other liquid. Now, our study proves that a sensor based on active electrolocation can identify the material type, which provides a reference for further engineering applications of active electrolocation in underwater identification. However, it is not perfect, which is fine in short-range object identification but long-range identification requires further improvement.

## Figures and Tables

**Figure 1 sensors-20-04608-f001:**
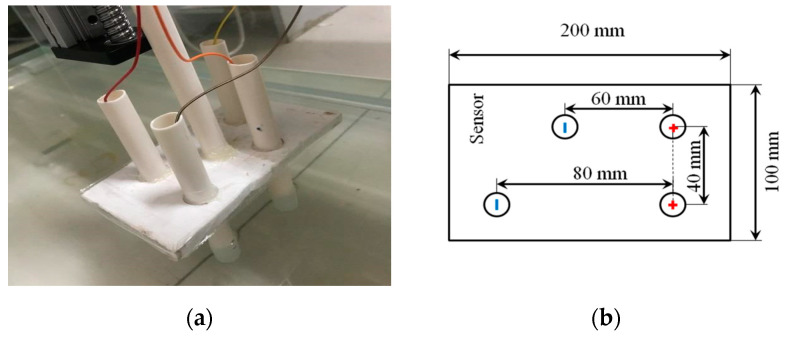
(**a**) The picture of the bionic sensor and (**b**) the structure of the bionic sensor.

**Figure 2 sensors-20-04608-f002:**
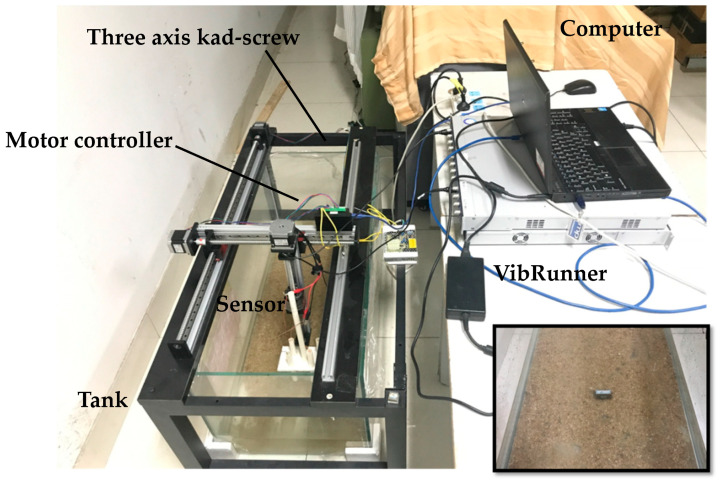
The active electrolocation test device: the sensor was attached to the tip of a stick whose planar motions were controlled by three-axis ball screw. The scene inside the water tank is shown in the lower right corner.

**Figure 3 sensors-20-04608-f003:**
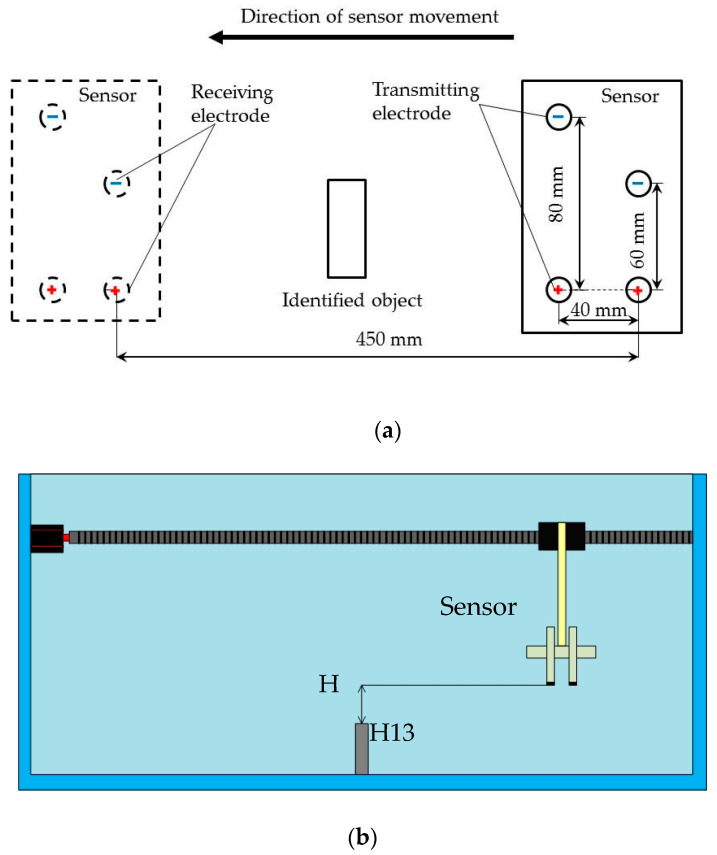
(**a**) Schematic of sensor identifying underwater objects; (**b**) Schematic of sensor’s movement underwater. H: Height of identification.

**Figure 4 sensors-20-04608-f004:**
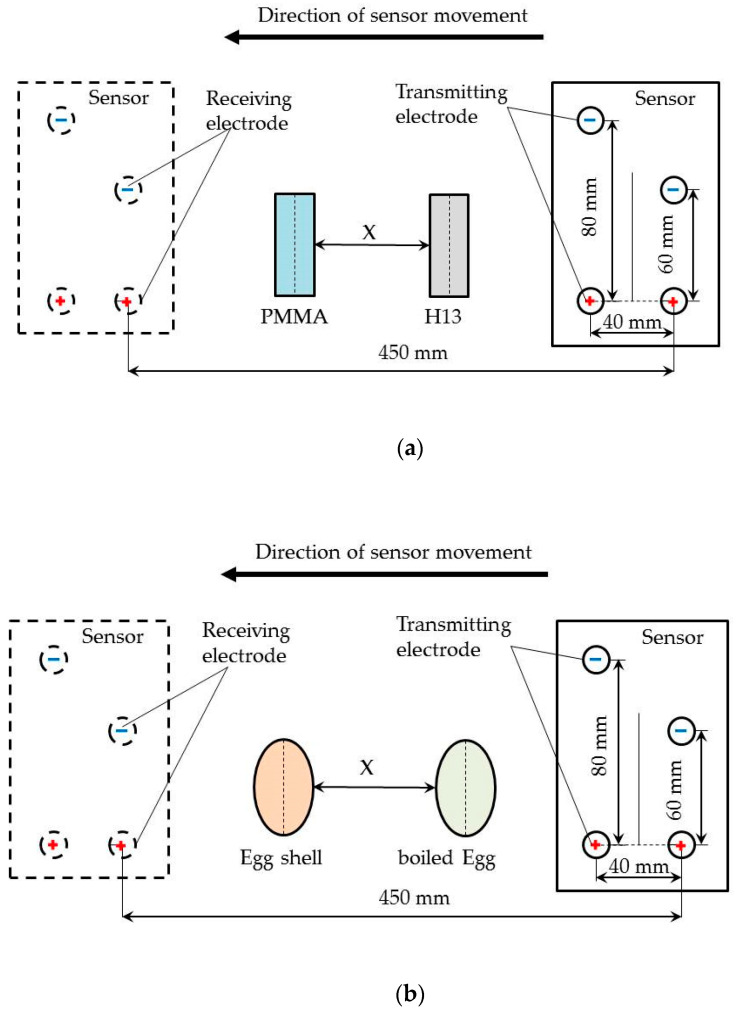
The schematic of the sensor identifying and locating two kinds of materials. The sensor passed above the identified objects along the X axis, (**a**) Schematic of the H13 and PMMA identification test; (**b**) Schematic of eggshell and boiled egg identification test.

**Figure 5 sensors-20-04608-f005:**
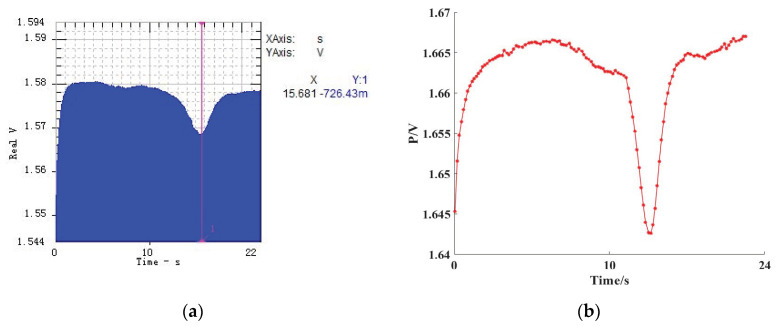
The comparison between the original image and the processed image, (**a**) The sine signal electric image saved from the SO Analyzer. (**b**) The image processed by Matlab.

**Figure 6 sensors-20-04608-f006:**
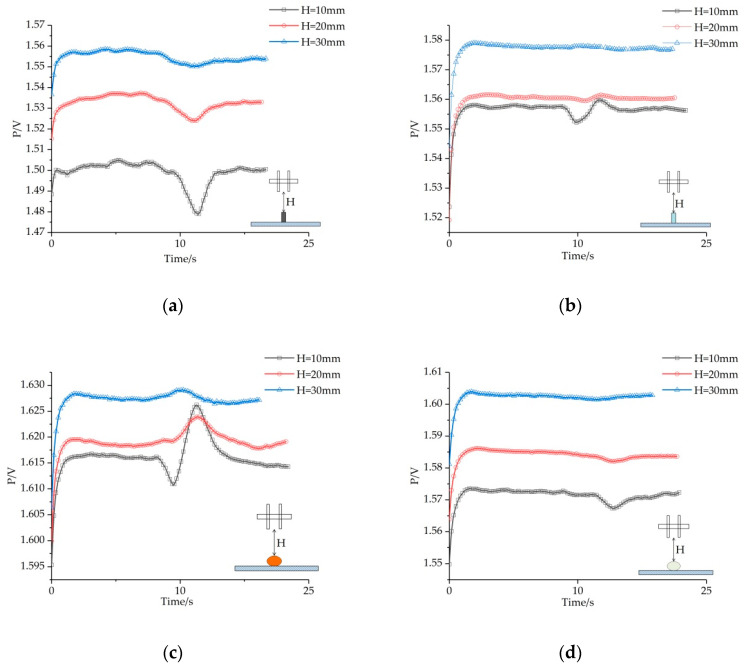
Effects of lift-off distance on identification performance. (**a**–**d**) Electric signal images evoked by objects of four different materials under same conditions. (**a**) Electric image of sensor identifying H13. (**b**) Electric image of sensor identifying the PMMA. (**c**) Electric image of sensor identifying the eggshell. (**d**) Electric image of sensor identifying the poached egg. (**e**) Electric image when there is no object. (**f**) Curve graph of the amplitude variation of the electrical signal of different materials.

**Figure 7 sensors-20-04608-f007:**
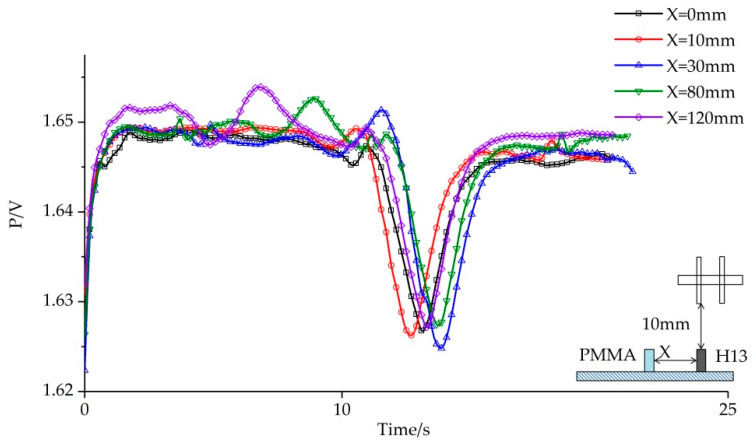
The electric signal image identifying PMMA and H13 under different values of “X”.

**Figure 8 sensors-20-04608-f008:**
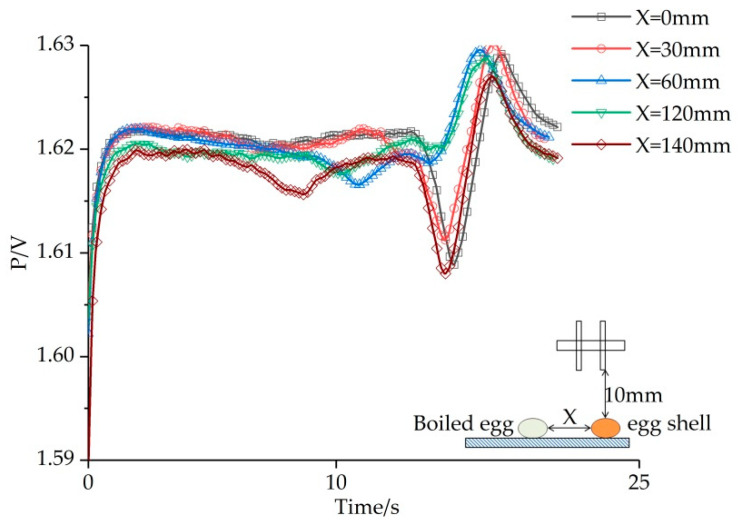
The electric image of identifying boiled egg and eggshell under different value of “X”.

**Figure 9 sensors-20-04608-f009:**
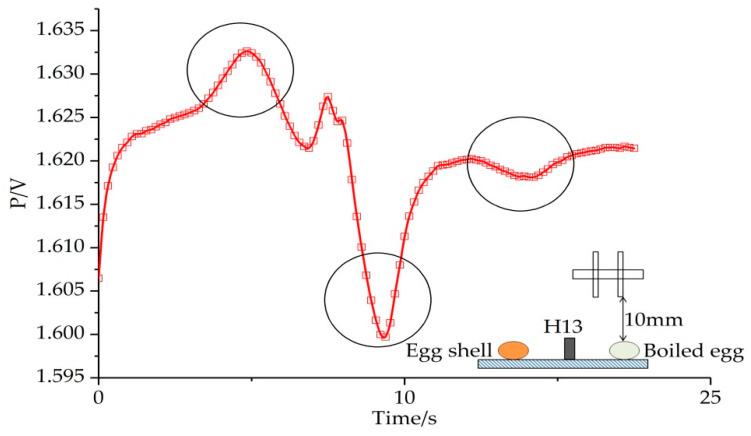
The electric image of identifying the boiled egg, H13, and eggshell. The three circles expressed the position of the poached egg, H13, and eggshell.

**Table 1 sensors-20-04608-t001:** The parameters of the experimental materials.

Materials	Size (mm^3^)	Conductivity (Siemens/m)
H13	24 000	2 × 10^6^
PMMA	24 000	0
Poached Egg	49 000	0.65
Eggshell	49 000	0

**Table 2 sensors-20-04608-t002:** Analysis table for positioning errors of the H13 and PMMA.

	X	0 mm	10 mm	30 mm	80 mm	120 mm

∆*t*(s)		--	--	1.71	4.242	6.16
*l*(mm)		--	--	34.4	84.84	123.2
δ		--	--	14.7%	6.05%	2.6%

**Table 3 sensors-20-04608-t003:** Analysis table for positioning errors of the boiled egg and eggshell.

		0 mm	30 mm	60 mm	120 mm	140 mm

∆*t*(s)		--	--	3.4	6.2	6.81
*l*(mm)		--	--	68	124	136.
δ		--	--	13.3%%	3.3%	2.7%

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
