# Peer review of "Research on the Recognition Performance of Bionic Sensors Based on Active Electrolocation for Different Materials"

_sensors, 2020, doi:10.3390/s20164608_

Round 1
Reviewer 1 Report
This paper designs and validates a bionic sensor based on active electrolocation for different materials.
The text needs a full grammar and typo review. Some issues include not including spaces between references and sentences, lack of standardization on author citation, text repetition (studied, studied). Also, the introduction should clearly present the paper contributions. Furthermore, the structure of the paper description is missing.
My main concerns with this paper are the lack of comparison with related work and weak experiments. First, it is not clear if other works mimic electric fish sensing capabilities. Second, I consider the experimental setup too simplistic and think the authors should carry out more work to present more consistent results with real-life environments. I suggest the authors perform tests in larger spaces, consider turbidity, analyze impacts on conductivity, and study the implications of using the sensor on a non-flat structure.
Reviewer 2 Report
The authors present an interesting and valuable contribution to underwater sensing using a fairly novel approach based on electrolocation. Overall, the work is justified and of interest to the readership of MDPI sensors. However, there are a few modifications and further clarification necessary before this reviewer can recommend it fully for publication:
1) Abstract: specify with physical units what the authors define as "close range", as this varies widely depending on the audience.
2) Introduction: similar to the abstract, the authors need to provide references and phyiscal dimensions for what they refer to as "long distance and large size".
3) Introduction: in general, a summary table of the literature with the paper, size, materials, types of organism / device would be very benefical to gain a better overview of the background work done on the electrolocation topic.
4) Introduction: the case study of the missing plane is interesting, but very out of scope with the actual materials and physical dimensions of the tested apparatus. The authors are therefore strongly encouraged to revise this section with a more practical example of their proposed device based on its performance.
5) Materials and Methods: The "shelled egg" can be revised as "egg shell" as this is a more common description.
6) Materials and Methods: What the exact number of replicates, and how much did they deviate from each other (e.g. provide standard deviation of H).
7) Materials and Methods: Why was 1000 Hz and 5 V chosen? This needs to be clearly justified and / or referenced accordingly.
8) Materials and Methods: Please cite the Matlab peak function more specifically.
9) Figure 6: What experimental signal is this?
10) Figure 7: Please put the name of each object in the panels to improve clarity.
11) Results and Discussion (ln 294-298): This statement is very general. Please revise and provide evidence / references to your own work which support the claim that "the sensor we made can identify different materials effectively".
12) Conclusions: In general, the conclusions section should also put your findings into context with the state of the art. Please include an overview of your work by comparing it to some of the literature you cited in the Introduction.
13) Conclusions: In your four-point summary, include the values of your performance and evaluation to support your written claims (e.g. "the shorter the lift-off distance" can be revised to include the actual distances used in this work).
Reviewer 3 Report
In this paper, a bionic sensor inspired by weakly electric fish has been made. To study its performance, an active electrolocation test device has been designed and buit. In this work, Firstly, the lift-off distance of identifying different materials has been optimized. Secondly, the influence of distance between two objects on positioning accuracy has been studied using the optimized lift-off distance. Finally, the identification tests have been conducted using three kinds of materials.The results are relevant, and I recommend this manuscript for publication in Sensors after some minor revisons.
General comments
- Please improve your abstract. It is not well-written.
- Please write your keyword in the alphabet.
- Please add the outline to the end of your introduction.
- Please add the outline of your paper “The rest of this study is organized as follows: … presented in Section 2. In Section 3, the …. are investigated and discussed and finally Summary and conclusions are presented in Sections 4.”
- Please increase the quality of the images and graphs.
-The paper contains way too many typos as well as grammatical mistakes and should be proofread carefully, possibly with the help of native speakers.
Round 2
Reviewer 1 Report
The authors have sufficiently addressed my comments.